# A Review of the Characteristics of Clinical Trials and Potential Medications for Alcohol Dependence: Data Analysis from ClinicalTrials.gov

**DOI:** 10.3390/medicina59061101

**Published:** 2023-06-07

**Authors:** Fahad S. Alshehri

**Affiliations:** Department of Pharmacology and Toxicology, College of Pharmacy, Umm Al-Qura University, Makkah 24382, Saudi Arabia; fsshehri@uqu.edu.sa

**Keywords:** alcohol dependence, clinical trials, alcoholism, alcohol use disorder, alcohol abuse

## Abstract

*Objective*. This study provides a comprehensive analysis of the characteristics of clinical trials related to alcohol dependence that are registered on ClinicalTrials.gov. *Methods*. All ClinicalTrials.gov trials registered up to 1 January 2023 were examined, focusing on trials that involved alcohol dependence. All 1295 trials were summarized by presenting their characteristics and results and reviewed most intervention drugs used in the treatment of alcohol dependence. *Results*. The study analysis identified a total of 1295 clinical trials registered on ClinicalTrials.gov that were focused on alcohol dependence. Of these, 766 trials had been completed, representing 59.15% of the total, while 230 trials were currently recruiting participants, accounting for 17.76% of the total. None of the trials had yet been approved for marketing. The majority of the studies included in this analysis were interventional studies (1145 trials, or 88.41%), which accounted for most of the patients enrolled in the trials. In contrast, observational studies represented only a small portion of the trials (150 studies, or 11.58%) and involved a smaller number of patients. In terms of geographic distribution, the majority of registered studies were located in North America (876 studies, or 67.64%), while only a small number of studies were registered in South America (7 studies, or 0.54%). *Conclusions*. The purpose of this review is to provide a basis for the treatment of alcohol dependence and prevention of its onset through an overview of clinical trials registered at ClinicalTrials.gov. It also offers essential information for future research to guide future studies.

## 1. Introduction

The global consumption of alcohol has a major effect on public health [1]. Alcohol is a depressant that affects the central nervous system and can impair cognitive function, memory, and judgment [2]. Therefore, alcohol consumption can increase accidents and injuries, including falls, motor vehicle accidents, and domestic violence [3]. The prolonged alcohol abuse can lead to a range of chronic diseases, including liver disease, cardiovascular disease, cancer, and neurological disorders [4]. Alcohol abuse is a leading cause of liver disease, including alcoholic hepatitis, cirrhosis, and liver cancer [5]. Alcohol abuse can also have serious implications for mental health [6]. Studies have linked excessive alcohol consumption to depression and anxiety, and in some cases, alcohol abuse can lead to addiction and substance abuse disorders [7,8].

There is no doubt that alcohol has profound implications for populations and public health [1]. It is widely believed to be the only psychoactive substance with an addictive potential that cannot be controlled internationally by legally binding regulatory frameworks, despite its significant impact on public health [8]. In recent years, there has been a significant increase in the number of studies conducted on the adverse effects of alcohol on health [9,10,11]. Several studies indicate that even moderate alcohol consumption can contribute to various acute and chronic health problems [4,12]. Alcohol consumption has been linked to more than 200 diseases, according to several studies, and chronic illnesses are affected by their pathogenicity and lethality, depending on their amount, quality, and consumption pattern [13].

In most treatment programs for alcoholism, patients are required to attend regular anonymous meetings of Alcoholics Anonymous groups. These meetings provide a supportive and non-judgmental environment where patients can share their experiences, receive guidance, and build relationships with others who are also struggling with addiction [14]. Relapse-prevention techniques are also used in many treatment programs to enable patients to learn how to prevent relapse once they have achieved abstinence [15]. Psychologists have developed therapeutic methods based on behavioral psychology, which have proven to be effective in the treatment of alcoholism [16]. One of the most popular forms of behavioral therapy is cognitive-behavioral therapy (CBT). CBT assumes that behavior is primarily learned, and that learning can provide solutions to behavioral problems. CBT can help patients develop coping skills, identify and challenge negative thoughts and beliefs that contribute to drinking, and learn new ways to manage stress and other triggers for alcohol use.

Pharmacological agents have also been used in the treatment of alcoholism. The medication disulfiram, which has been in use since the late 1940s, works by causing unpleasant side effects such as nausea and vomiting when alcohol is consumed [17]. This can help patients develop an aversion to alcohol and discourage them from drinking. Naltrexone is another medication that has been approved by the FDA for the treatment of alcoholism, based on randomized clinical trials that showed its effectiveness in reducing alcohol cravings and the likelihood of relapse [18]. Despite the positive results of pharmacotherapy for improving treatment outcomes, naltrexone has not been widely used, and there is still a need for further research into the effectiveness of pharmacological treatments for alcoholism [19].

The purpose of this article is to provide a comprehensive overview of the current state of pharmacological treatment research for alcohol dependence, including recent clinical trials and key characteristics of this research field. Based on a thorough analysis of 1295 clinical trials related to alcohol dependence, focusing specifically on the most extensively studied drugs that completed phase 4 interventional studies, this study provides valuable insights into the effectiveness of drugs for treating alcohol dependence. The goal is to contribute to the ongoing efforts to develop more effective pharmacological treatments for alcohol dependence, which remains a significant public health concern globally.

## 2. Methods

### 2.1. Data Sources and Search

The ClinicalTrials.gov database, which serves as a comprehensive registry of clinical trials worldwide, was carefully examined to identify all clinical trials that pertain to alcohol dependence, and this search was conducted up until 1 January 2023. Following this extensive search, a detailed summary of the retrieved information was compiled, which is presented in Table 1. This summary outlines the key characteristics of each clinical trial, such as the study type, the region where the study was conducted, the number of patients enrolled, the study status, and the phase of the study. Our objective was to provide a comprehensive overview of the clinical trials that have been conducted for the treatment of alcohol dependence, in order to facilitate future research in this area and to help guide the development of more effective treatments. The study design followed the 2020 Preferred Reporting Items for Systematic Reviews and Meta-Analyses (PRISMA) guidelines. The full search strategy and inclusion/exclusion criteria are available upon request.

### 2.2. Data Extraction, Collection, and Analysis

The information for this study was obtained from ClinicalTrials.gov and organized into a table (Table 2). The following categories were used to classify each study: study status, number of participants, study type, region, and study phase. Clinical trials for alcohol dependence were conducted on various continents, including Africa, East Asia, Europe, North America, Asia, Pacifica, South America, and other regions that were unidentified. Only studies with reported locations were included in the counts for each region. The status of each study was classified into different categories such as recruiting, completed, terminated, and more. The number of participants was categorized into three groups: less than 1000, 1000 to 5000, and above 5000. The study types included in this study were categorized into two main groups: interventional and observational. Interventional studies involve administering a treatment or medication to participants, while observational studies involve observing and collecting data from participants without administering any new treatment. In addition, the study phase of each trial was also categorized and analyzed. The phases ranged from early phase 1 trials, which involve testing a new medication or treatment in a small group of healthy individuals, to phase 4 trials, which are conducted after a medication has been approved for use and are focused on monitoring the long-term effects of the medication in a larger population. To identify the medications examined in this study, completed phase 4 interventional studies with reported results were analyzed and selected, as illustrated in Figure 1. These medications were then evaluated based on their efficacy and safety in treating alcohol dependence. Statistical analysis was conducted using Excel, and the data were expressed as absolute numbers and percentages for categorical variables.

The inclusion criteria for the study involved selecting clinical trials related to alcohol dependence, specifically focusing on drugs that had completed phase 4 studies and were of the interventional type. The aim was to identify drugs that have been extensively studied in the context of alcohol dependence treatment. The screening process aimed to filter out trials that did not meet these criteria, ultimately identifying a subset of drugs that were studied more frequently than others. This subset provided valuable insights into which drugs have been the most effective in treating alcohol dependence. The exclusion criteria involved excluding clinical trials not related to alcohol dependence, drugs that had not completed phase 4 studies, and non-interventional studies that did not provide data on the effectiveness of drugs in treating alcohol dependence. By applying these criteria, the study aimed to focus on the most relevant and informative data from the available pool of clinical trials. This allowed for a clear and concise presentation of the findings and helped to identify any trends or patterns in the data. By analyzing the characteristics of clinical trials and medications for alcohol dependence, this study aims to contribute to the development of more effective treatments and interventions for individuals struggling with alcohol use disorders.

## 3. Results

### 3.1. Clinical Trials Characteristics

This study aimed to investigate the characteristics of clinical trials registered on alcohol dependence in the ClinicalTrials.gov database. The search yielded a total of 1295 registered clinical trials on alcohol dependence as of 1 January 2023. Table 2 provides an overview of the trial characteristics, including status and the number of patients enrolled, as well as the clinical trial study types. Of the these studies registered clinical trials, 766 (59.15%) were completed, 230 (17.76%) were in the process of recruiting, and none of these clinical trials had been approved for the market. The majority of the clinical trials (88.41%) were of the interventional study type (*n* = 1145), while only 150 (11.58%) were of the observational study type. The number of enrolled patients was classified into three categories: less than 1000 (*n* = 1224, 94.51%), 1000–5000 (*n* = 57, 4.40%), and more than 5000 (*n* = 14, 1.08%). The distribution of registered studies was worldwide, with the highest number of trials registered in North America (*n* = 876, 67.64%) and the lowest in South America (n = 7, 0.54%).

### 3.2. Most Studied Drugs in the Treatment of Alcohol Dependence

These findings provide insight into the current landscape of clinical trials registered on alcohol dependence and highlight the need for increased attention and resources to address this global health issue. There have been numerous drugs studied for the treatment of alcohol dependence, but some have been studied more extensively than others. After conducting a thorough analysis of the data collected from ClinicalTrials.gov, the study identified a significant number of clinical trials related to alcohol dependence, totaling 1295 studies. Given the large number of studies available, the results were then filtered to focus on the most studied drugs for this condition. Thus, the study focused specifically on drugs that had completed phase 4 studies and were of the interventional type (Table 3). The screening process allowed us to identify a subset of drugs that were studied more frequently than others, providing us with valuable insights into which drugs have been the most effective in treating alcohol dependence. Figure 1 helps illustrate the findings in a clear and concise manner.

### 3.3. Naltrexone

According to ClinicalTrials.gov, naltrexone has been studied in 101 studies on alcohol dependance and 115 studies on alcohol-related conditions. Naltrexone is a long-acting opioid receptor antagonist approved by the Food and Drug Administration for treating opioid dependence and alcohol dependence [20]. It has been demonstrated that naltrexone and its active metabolite have reversible competitive antagonist activities at μ-opioid and κ-opioid receptors [21], with the lowest affinity for κ-opioid receptors and the highest affinity for μ-opioid receptors [22]. These mechanisms are consistent with naltrexone’s neuropharmacological profile as a competitive antagonist of the μ-opioid receptor [23]. The activation of the ventral tegmental area may indirectly facilitate the release of dopamine in the nucleus accumbens through GABAergic and dopaminergic receptors pathway [24,25]. Thus, upon consumption of alcohol, endogenous opioids are released, which activates the opioid receptor, contributing to the hedonic effects of alcohol [26,27]. Conversely, naltrexone blocking μ-opioid receptor delays alcohol-related dopamine release in the nucleus accumbens, thus reducing alcohol reward and self-administration [28]. Preclinical studies have shown that naltrexone reduces alcohol consumption and seeking behavior in rodents and primates. Animal research studies have initially suggested that naltrexone blocks ethanol’s positive reinforcing effects when used to treat alcohol use disorders [29,30,31]. It has been reported that naltrexone reduced the amount of ethanol administration in rats, as well as the duration of time spent in a state of ethanol intoxication [32]. Additionally, Naltrexone has been shown to reduce alcohol consumption in non-human primates, with a study on rhesus macaques demonstrating a reduction in alcohol intake following administration of the drug [33]. The blockade of opioid receptors was reported to reduce ethanol consumption behaviors in various species and reduce human self-administered ethanol consumption [34]. The results of these studies led to the development of clinical trials involving naltrexone for the treatment of alcohol dependence in humans [35,36,37].

### 3.4. Topiramate

According to ClinicalTrials.gov, topiramate has been studied in 15 studies on alcohol dependance and 29 studies on alcohol-related conditions. The topiramate structure contains a monosaccharide chemical structure containing sulfamate [38]. Despite its structural differences from other antiepileptic drugs, topiramate has different pharmacological properties that contribute to its anticonvulsant properties and its therapeutic effects on other neuropsychiatric disorders [39]. By blocking state-dependent sodium channels, it inhibits actions associated with high and repetitive cellular discharges [40]. Topiramate potentiates GABA activity by inducing a chloride ion flux in the neurons [41]. It is likely that topiramate binds to a non-benzodiazepine receptor site on GABA-A receptors since the benzodiazepine antagonist flumazenil does not inhibit this action [42]. Furthermore, topiramate antagonizes the AMPA/kainate glutamate receptors, reducing neuronal excitability [43]. Topiramate has been shown to be effective in treating alcohol dependence in clinical practice [44]. Additionally, topiramate has been shown to reduce symptoms of anxiety and depression, which are common co-occurring conditions in people with alcohol use disorder [45]. Studies have compared the efficacy of topiramate with two drugs used for alcohol dependance, disulfiram and naltrexone. According to these studies, topiramate may be more effective than standard doses of oral naltrexone. One 12-week randomized, double-blind, placebo-controlled trial found that topiramate showed superior outcome compared to naltrexone, although the study was underpowered [46]. In addition, two 6-month randomized, open-label trials found that topiramate at a mean dose of 200 mg/day was better than naltrexone at reducing alcohol intake and craving [47]. In addition to improving craving-priming and dependence intensity scales, topiramate led to a significant decrease in transaminase levels and the number of drinking days per month reported [48]. Based on recent reports, its use appears to be safe and well tolerated and can be considered as a potential medication for alcohol dependence to reduce the reinforcement effects associated with alcohol consumption treatment.

### 3.5. Acamprosate

According to ClinicalTrials.gov, acamprosate has been studied in 12 studies on alcohol dependance and 26 studies on alcohol-related conditions. Acamprosate is an N-acetylated calcium salt containing sulfur amino acid that resembles both GABA and glutamate in structure [49]. Studies have reported that acamprosate can decrease ethanol consumption in rodents that have been exposed to ethanol for a prolonged period and developed ethanol dependence [50,51]. Research on rats suggested that it reduced the increased ethanol consumption associated with abstinence from ethanol [52]. Acamprosate has been shown to attenuate alcohol withdrawal behavioral associated with neurochemical symptoms [53]. For instance, acamprosate administration was found to reduce hyperactivity and glutamate levels during withdrawal from ethanol; this effect can observe if acamprosate was given in the initial stage of withdrawal [54]. From the results of these studies, it is evident that acamprosate can modify the consumption of ethanol. Few studies have revealed that acamprosate affects the transmission of NMDA receptors and GABA A receptors. Thus, acamprosate can decrease glutamate levels during alcohol withdrawal, increase endorphin levels in individuals with high alcohol intakes, and influence the hypothalamic-pituitary-adrenal axis. Pharmacologically, acamprosate targets subclinical withdrawal symptoms and even provides neuroprotection [55]. Acamprosate may treat alcohol dependence since it affects glutamate, NMDA, and GABAA transmission. It has been reported that acamprosate compensates for neurobiological derangements induced by alcohol withdrawal and potentiates GABA’s effects on GABAA receptors while decreasing NMDA, AMPA, and kainate receptor function [56].

### 3.6. Baclofen

According to ClinicalTrials.gov, baclofen has been studied in 9 studies on alcohol dependance and 21 studies on alcohol-related conditions. Baclofen, a gamma-aminobutyric acid (GABA)-B receptors agonist, relieves the spasticity of muscles by inhibiting the transmission of monosynaptic and polysynaptic reflexes in the spinal cord [57]. In preclinical studies, GABA-B receptors are implicated in memory storage [58], reward [59], motivation [60], and anxiety [61]. It has been shown that activation of GABA-B receptors by baclofen may produce neuroprotective anti-inflammatory effects that may be relevant to treating alcohol dependence [62,63,64]. Several preclinical studies have shown that baclofen inhibits alcohol consumption in rats [32,65,66]. It is important to note that alcohol and baclofen do not activate the same receptors. The effects of baclofen on GABA-B receptors are selective [67], and alcohol does not directly affect these receptors [68]. Both substances may, however, indirectly act on the same systems, particularly the glutamatergic and GABAergic systems [32]. Baclofen partially reverses the effects of alcohol on GABA-B, although alcohol does not directly affect GABA-B [69]. A number of studies have been conducted on the effects of baclofen on animal models for alcoholism [70,71]. It has been reported that baclofen administered systemically can decrease alcohol consumption acquisition and maintenance, motivation to drink, relapse-like drinking, alcohol withdrawal symptoms, and cue-induced reinstatement [65,72,73,74]. Furthermore, research has demonstrated that baclofen enantiomer (R) reduces alcohol self-administration in rats, thus supporting the claim that baclofen reduces alcohol consumption [75]. Moreover, baclofen has been shown to have a positive effect on alcohol-related behaviors such as preference and consumption, further confirming its potential as a treatment for alcohol use disorder. Therefore, baclofen appears to have the potential to reduce alcohol consumption in humans by influencing the underlying biological systems.

### 3.7. Ondansetron

According to ClinicalTrials.gov, ondansetron has been studied in four studies on alcohol dependance and fifteen studies on alcohol-related conditions. Ondansetron functions as a 5-HT3 receptor antagonist that is important in regulating the severity of alcohol consumption [76,77]. Ondansetron has the potential to treat alcohol dependence, but its effectiveness appears to be confined to alcoholics with a genetic inclination towards excessive drinking [78,79]. It is believed that ondansetron produces an overall suppression of dopamine signaling in mesolimbic brain regions by antagonizing 5-HT3 receptors in cortico-mesolimbic [80]. Thus, the serotonin receptors regulated dopamine outflow in the rat nucleus accumbens [81]. By inhibiting the serotonin transporter (5-HT), ondansetron is believed to decrease the alcohol reward by modulation of the cortico-mesolimbic dopamine system [82]. Several studies have shown that ondansetron may help treat alcohol dependence [83,84,85]. Several studies have demonstrated that ondansetron reduces drinking in both humans and animals [85,86,87]. Compared with individuals with late-onset alcoholism, ondansetron can reduce drinks per drinking day and increase abstinence days in people with early-onset alcoholism [86]. In humans, it has been shown that pretreatment with ondansetron, a selective 5-HT3 receptor antagonist, attenuates low-dose alcohol-induced subjective effects and the desire to drink [88]. Nevertheless, ondansetron and alcohol can also have both stimulant and sedative effects, which may reduce their functionality in alcohol dependence [89,90]. The use of ondansetron outside of research settings is not currently feasible due to the fact that it is not yet available on a commercial basis at the therapeutic dose necessary for alcohol dependence treatment.

### 3.8. Zonisamide

According to ClinicalTrials.gov, zonisamide has been studied in four studies on alcohol dependance and eleven studies on alcohol-related conditions. Zonisamide is an anticonvulsant drug belonging to the methanesulfonamide group, which shares a sulfamoyl group with acetazolamide, an analog of an aryl sulfonamide [91]. There is still a great deal to be learned about the neuronal mechanisms through which zonisamide reduces alcohol consumption [92,93]. One possible mechanism is that zonisamide may modulate alcohol-drinking behavior by inhibiting the excitability of AMPA, N-methyl-d-aspartate, and/or kainate receptors in the hippocampus and frontal and anterior cingulate cortex [94,95]. It has been suggested that zonisamide may suppress alcohol-induced brain excitability through its positive modulatory interaction with GABA A receptors [96] and its antagonism of kainate receptors containing the GluK1 subunit [97]. According to recent studies, sensitivity to zonisamide-induced reductions in heavy drinking is associated with a specific polymorphism in the kainate receptor GluK1 subunit gene [85,98]. Therefore, zonisamide inhibits glutamate’s release and decreases glutamate’s excitatory postsynaptic release through postsynaptic mechanisms. In addition to enhancing the activity of GABA receptor systems, zonisamide may also reduce brain excitability by down-regulating GABA transporter proteins in the brain. A study conducted on rats found that treatment with zonisamide resulted in a significant reduction in ethanol consumption, as well as a differential influence on memory [99]. Additionally, another study conducted on mice found that zonisamide treatment led to a decrease in ethanol-induced conditioned place preference, suggesting a potential for the medication to reduce the rewarding effects of ethanol [92]. These preclinical findings suggest that zonisamide may be a promising candidate for the treatment of ethanol dependence in humans and warrant further investigation in clinical trials.

### 3.9. Quetiapine

According to ClinicalTrials.gov, quetiapine has been studied in three studies on alcohol dependance and twelve studies on alcohol-related conditions. It has been reported that chronic administration of quetiapine reduced ethanol consumption in rats, indicating a potential role in the treatment of alcohol use disorders [100]. Quetiapine binds to multiple receptors, including K 1/2 and K 1/2, dopamine D1 and D2, histamine H1, and adrenergic receptors a1 and a2 [101]. Understanding quetiapine’s neurobiological effects and targets is essential to understanding its clinical effects on alcoholism [102]. To understand quetiapine’s effects on alcoholism, understanding its impact on serotonergic and dopaminergic activity may be particularly relevant. Thus, it is believed that quetiapine works as 5-HT2A antagonist receptors on the dopaminergic neurons in the mesolimbic and mesocortical areas of the brain [103]. Besides blocking 5-HT2A receptors, quetiapine may also antagonize D2 receptors [104]. Compared to other antipsychotics, quetiapine is less likely to induce addiction or dependence, unlike other dopamine antagonists [105], due to its lower dopamine blockade potency [106]. This may also explain quetiapine’s better tolerability profile since 5-HT2A receptor affinities are higher than D2 receptors in atypical antipsychotic medications [107]. Specifically, 5-HT2A and D2 antagonists on mesolimbic dopaminergic pathways into the nucleus accumbens cause a reduction in dopaminergic neurotransmission into those regions and modulate dopaminergic neurons’ activity [100]. Despite the antagonistic effects of D2 receptors, it remains unclear which neurobiological mechanism atypical antipsychotics target to relieve psychotic symptoms. It is possible to understand quetiapine’s clinical effects in the treatment of alcoholism through its antagonistic effects on 5-HT2A and D2, which decrease dopaminergic activity in the mesolimbic pathway. Several preclinical studies have explored the potential effects of quetiapine in ethanol dependence in animals [108,109]. Additionally, quetiapine treatment decreased anxiety-like behavior and increased social interaction in ethanol-dependent rats, which are commonly observed symptoms in alcohol dependence [110]. Furthermore, quetiapine has also been shown to modulate dopamine and glutamate neurotransmission, which are implicated in the neurobiology of addiction [111,112]. Overall, these preclinical findings suggest that quetiapine may have potential therapeutic effects in ethanol dependence, although further research is needed to determine its clinical efficacy and safety in humans.

### 3.10. Dutasteride

According to ClinicalTrials.gov, dutasteride has been studied in two studies on alcohol dependance and five studies on alcohol-related conditions. Dutasteride is a second 5-alpha reductase enzyme inhibitor approved by the FDA for treating benign prostatic hyperplasia [113]. It is believed that dutasteride is an effective treatment option for patients with mild to severe symptomatic benign prostatic hyperplasia [114,115] and may be able to reduce the chance of developing prostate cancer [116]. In addition, there has been substantial evidence that dutasteride effectively reduces the risk of prostate cancer and delays the progression of the disease in men [117,118,119]. On the other hand, dutasteride has been reported to reduce alcohol consumption during the first two weeks after taking the drug by inhibiting the 5-alpha reductase enzyme, which reduces the reinforcing effects of alcohol [120]. Despite this, it has been suggested that reducing alcohol use caused by dutasteride in alcohol-treated subjects may increase alcohol drinking in other ways that were not investigated by the study [121]. Unfortunately, little is known about dutasteride in alcohol treatment since it has not been studied extensively yet. Future studies are still warranted to investigate its role in alcohol drinking and alcohol dependency.

### 3.11. N-Acetylcysteine

According to ClinicalTrials.gov, N-acetylcysteine (NAC) has been studied in one study on alcohol dependance and four studies on alcohol-related conditions. NAC is a sulfur-containing compound derivative of the amino acid L-cysteine, a precursor for glutathione synthesis that increases intracellular glutathione levels at the cellular level and stimulates cytosolic enzymes responsible for glutathione regeneration [122]. It is used therapeutically as a mucolytic agent and has shown promise in treating several psychiatric disorders, including addiction [123,124]. It is believed that NAC is involved in the modulation of cysteine/glutamate antiporters in astrocytes and the reset of glutamatergic neurotransmission through the tonic activation of extrasynaptic mGluR2/3 receptors [125]. It has been reported that NAC can attenuate the reinforcing effects of several addictive drugs [126]. For example, it has been reported NAC treatment prevents behavioral sensitizations associated with cocaine and ethanol administration, which is particularly noteworthy since behavioral sensitization serves as a paradigm for studying the changes in plasticity following repeated exposure to drugs [127,128]. Furthermore, it has been demonstrated that chronic alcohol consumption results in neuroinflammation and oxidative stress in the brain, which are highly interconnected proinflammatory and inflammatory cytokines processes [129,130,131]. NAC exerts neuroprotective effects by limiting glutamatergic activity, decreasing pro-inflammatory cytokines, and exerting antioxidant properties against alcohol or alcohol cessation-induced injuries [132]. Therefore, NAC has been reported to inhibit ethanol consumption and reduce the reward pathways associated with alcohol consumption motivation, craving, and relapse after abstinence [132,133,134]. Finally, as a natural anti-inflammatory and antioxidant, n-acetylcysteine can help reduce alcohol consumption to reduce oxidative stress caused by alcohol consumption and reduce the risk of developing alcohol dependence.

## 4. Limitations

ClinicalTrials.gov does not include all alcohol dependence trials performed worldwide. The study focused on ClinicalTrials.gov because of the tools available for its characterization. ClinicalTrials.gov is a comprehensive database of clinical trials and is the largest publicly available source of information on clinical trials. However, due to its focus on US-based trials, it does not include all trials that are being conducted worldwide. As such, it may not be comprehensive enough to capture all the trials related to alcohol dependence. In addition, as a result of the methodology of this study, it was not possible to include other drugs used to treat alcohol dependence.

## 5. Conclusions

The current study provides valuable insights into the landscape of clinical trials registered on alcohol dependence in the ClinicalTrials.gov database. The findings highlight a significant number of ongoing and completed clinical trials on this global health issue. The majority are interventional studies. Naltrexone, topiramate, and acamprosate were identified as the most studied drugs for alcohol dependence. The neuropharmacological mechanisms underlying their therapeutic effects were discussed. The results of this study emphasize the need for increased attention and resources toward developing effective treatments for alcohol dependence, given its significant global burden. These findings could aid researchers and clinicians in making informed decisions about selecting treatment options and improving outcomes for individuals affected by alcohol dependence.

## Figures and Tables

**Figure 1 medicina-59-01101-f001:**
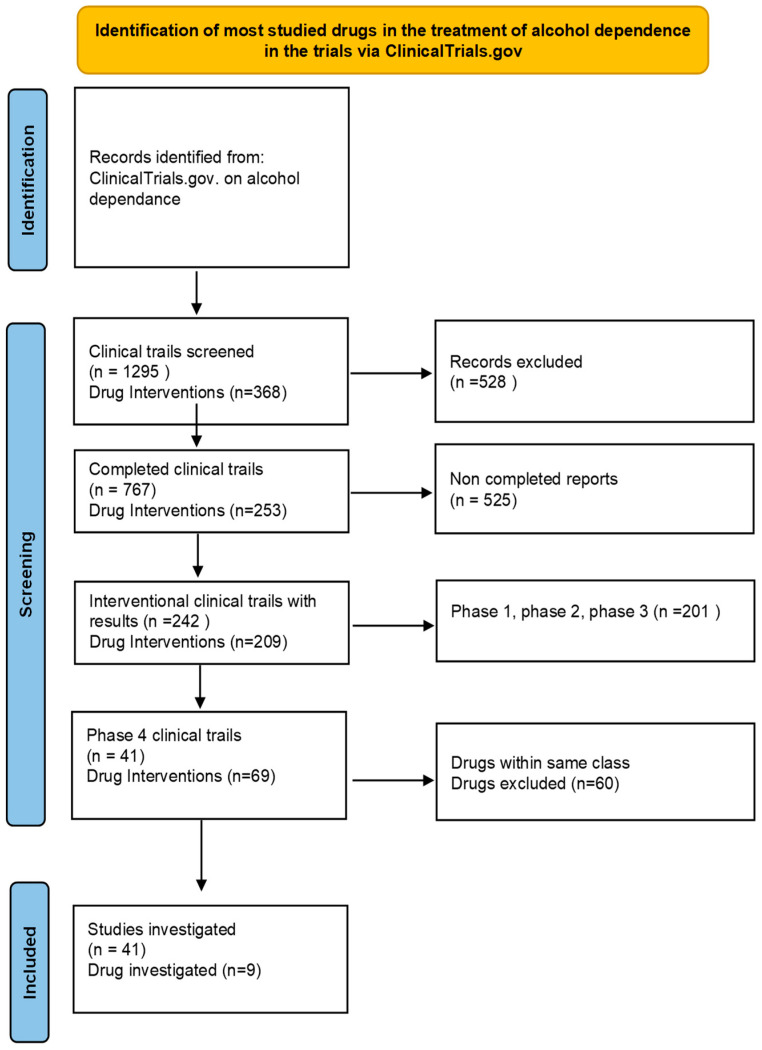
Methods used to identify most studied drugs in the treatment of alcohol dependence.

**Table 1 medicina-59-01101-t001:** Summary of the search.

Terms	Search Results	Entire Database
**Synonyms**		
**Alcohol Dependence**	1293 studies	1293 studies
**Alcoholism**	1276 studies	1276 studies
**Alcohol addiction**	27 studies	27 studies
**Alcohol dependency**	11 studies	11 studies
**Alcohol problem drinking**	2 studies	2 studies
**Chronic alcohol abuse**	1 study	1 study
**Dependence**	488 studies	2286 studies
**Dependency**	12 studies	125 studies
**Alcohol**	1295 studies	2609 studies
**Absolute ethanol**	--	1 study
**Ethanol measurement**	--	1 study

**Table 2 medicina-59-01101-t002:** Clinical trials characteristics.

Status	Number	Percentage
Recruiting	230	17.76%
Not yet recruiting	62	4.78%
Enrolling by invitation	12	0.92%
Completed	766	59.15%
Terminated	55	4.24%
Withdrawn	38	2.93%
Active, not recruiting	39	3.01%
Approved for market	0	0%
Suspended	5	0.38%
Unknown	93	7.18%
Number of patients Enrolled	Number	Percentage
<1000	1224	94.51%
1000–5000	57	4.40%
>5000	14	1.08%
Study type	Number	Percentage
Interventional (Clinical Trial)	1145	88.41%
Observational	150	11.58%
Region	Number	Percentage
World	1295	100%
Africa	16	1.23%
East Asia	18	1.38%
Europe	252	19.45%
North America	876	67.64%
Asia	37	2.85%
Pacifica	14	1.08%
South America	7	0.54%
Others	75	5.79%
Study Phase	Number	Percentage
Early Phase 1	24	1.85%
Phase 1	109	8.41%
Phase 2	317	24.47%
Phase 3	114	8.80%
Phase 4	122	9.42%
Not applicable (Describes trials without FDA-defined phases, including trials of devices or behavioral interventions).	531	41%

**Table 3 medicina-59-01101-t003:** Most studied drugs in the treatment of alcohol dependence in the clinical trials.

Drug Intervention Class	Name of Drugs	Number of Studies on Alcohol Dependence	Number of All Studies on Alcohol Related Conditions *
Analgesics	Naltrexone	101	115
Anticonvulsant	Topiramate	15	29
Alcohol Deterrents	Acamprosate	12	26
Muscle relaxant	Baclofen	9	21
Antiemetic	Ondansetron	4	15
Channel blocker	Zonisamide	4	11
Central nervous system depressant	Quetiapine Fumarate	3	12
5-alpha reductase inhibitors	Dutasteride	2	5
Amino acid	N-acetylcysteine	1	4

* Alcoholism, alcohol abuse, alcohol withdrawal, and alcohol use disorder.

## Data Availability

Not applicable.

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
