# Peer review of "A Review of the Characteristics of Clinical Trials and Potential Medications for Alcohol Dependence: Data Analysis from ClinicalTrials.gov"

_medicina, 2023, doi:10.3390/medicina59061101_

Round 1
Reviewer 1 Report
The review presented is unique so I understand the challenge the authors had in presenting the findings. I think the authors have to do make some major changes to make the findings clearer. My comments and suggestions are as follows, I hope it helps the authors.
1. The authors have not laid out the rationale very well. As a reader I was left wondering why is there a need for this review. I recommend the authors lay the rationale and aims of the review in more detail in the Introduction section. I found the aims outlined in the methods and results sections, I suggest authors to amend that to include in the Introduction.
2. I suggest that the authors move the last line in the first paragraph of the Results section to the Discussion section.
3. You have reported the total number of studies (1295) identified twice, within the first two paragraphs in Results. I think there is no need to repeat it.
4. I am not sure if it is the way the authors have formatted the figure but texts in Diagram 1 were cut off in my copy of the manuscript so I couldn't review them completely.
5. I don't think the description for each drug belongs in the Results section; authors should report here the names of the drugs studied, in how many studies etc (exactly what has been reported in Table 3).
6. The reference list needs to be edited, some references are incomplete or not incorrect.
Author Response
The review presented is unique so I understand the challenge the authors had in presenting the findings. I think the authors have to do make some major changes to make the findings clearer. My comments and suggestions are as follows, I hope it helps the authors.
I would like to thank the reviewer for these helpful comments, I have highlighted every change in red colour.
1. The authors have not laid out the rationale very well. As a reader I was left wondering why is there a need for this review. I recommend the authors lay the rationale and aims of the review in more detail in the Introduction section. I found the aims outlined in the methods and results sections, I suggest authors to amend that to include in the Introduction.
I would like to thank the reviewer, the following has been added “The purpose of this article is to provide a comprehensive overview of the current state of pharmacological treatment research for alcohol dependence, including recent clinical trials and key characteristics of this research field. Based on a thorough analysis of 1295 clinical trials related to alcohol dependence, focusing specifically on the most extensively studied drugs that completed phase 4 interventional studies, this study provides valuable insights into the effectiveness of drugs for treating alcohol dependence. The goal is to contribute to the ongoing efforts to develop more effective pharmacological treatments for alcohol dependence, which remains a significant public health concern globally”
2. I suggest that the authors move the last line in the first paragraph of the Results section to the Discussion section.
I would like to thank the reviewer .The suggestion has been performed.
3. You have reported the total number of studies (1295) identified twice, within the first two paragraphs in Results. I think there is no need to repeat it.
I would like to thank the reviewer. The suggestion has been performed.
4. I am not sure if it is the way the authors have formatted the figure but texts in Diagram 1 were cut off in my copy of the manuscript so I couldn't review them completely.
I have included a revised version of the manuscript.
5. I don't think the description for each drug belongs in the Results section; authors should report here the names of the drugs studied, in how many studies etc (exactly what has been reported in Table 3).
I would like to thank the reviewer. The suggestion has been performed for each drug
6. The reference list needs to be edited, some references are incomplete or not incorrect.
I would like to thank the reviewer. The references have been updated

Reviewer 2 Report
Review of manuscript entilted: “A Review of the Characteristics of Clinical trials and Potential Medications for Alcohol Dependence: Data analysis from Clinicaltrials.gov” authored by Fahad S Alshehri.
First of all I want to thank you for opportunity to review this interesting manuscript.
In the presented article, author collected information from clinicaltrials.gov to investigate, which pharmacological agents are most commonly used for pharmacotherapy of alcohol dependent patients.
Overall, I find this manuscript interesting, well-written and important due to worldwide problem of alcohol dependence. I provide some remarks below.
Major concerns:
- Author should consider adding some pie charts or share charts to present the frequencies instead of putting them in table. This is my personal point of view and can be ignored
- In conclusion, author writes “However, their effectiveness varies depending on individual factors, such as the severity of alcohol dependence, genetic factors, and co-occurring mental health conditions.”. I wonder if author performed any analysis regarding effectiveness of the pharmacotherapy with particular agents, this could be a good addition to your work and would greatly increase the impact of this research.
- Diagram
- Some parts of the text are cut and thus unreadable
- I do not understand last step “Included”, where author put 41 studies. Where were those 41 studies used, because I do not see any analysis matching this criterion
Minor concerns:
- Abstract
- Authors writes “we”, while he is the only author of the manuscript
- Results – author describes only location and type of studies; author does not mention drugs used for pharmacotherapy
- Conclusions – author writes about purpose of the study not conclusions
- Table 1
- “Search results*”, what does symbol mean at the end of this phrase?
- “studies” is plural form, author should correct this everywhere where he writes “1 studies” to “1 study”
- Table 3
- Please unify the font
Author Response
First of all I want to thank you for opportunity to review this interesting manuscript.
In the presented article, author collected information from clinicaltrials.gov to investigate, which pharmacological agents are most commonly used for pharmacotherapy of alcohol dependent patients.
Overall, I find this manuscript interesting, well-written and important due to worldwide problem of alcohol dependence. I provide some remarks below.
Major concerns:
Author should consider adding some pie charts or share charts to present the frequencies instead of putting them in table. This is my personal point of view and can be ignored
I would like to thank the reviewer for this suggestion. However, I think tables are more informative to reader.
In conclusion, author writes “However, their effectiveness varies depending on individual factors, such as the severity of alcohol dependence, genetic factors, and co-occurring mental health conditions.”. I wonder if author performed any analysis regarding effectiveness of the pharmacotherapy with particular agents, this could be a good addition to your work and would greatly increase the impact of this research.
I would like to thank the reviewer for this suggestion. The conclusion has been modified.
Diagram
Some parts of the text are cut and thus unreadable
I have included a revised version of the manuscript.
I do not understand last step “Included”, where author put 41 studies. Where were those 41 studies used, because I do not see any analysis matching this criterion
I would like to thank the reviewer for this comment. I have included in the method section as suggested.
Minor concerns:
- Abstract
Authors writes “we”, while he is the only author of the manuscript
I have included a revised version of the abstract as requested.
Results – author describes only location and type of studies; author does not mention drugs used for pharmacotherapy
I would like to thank the reviewer for this comment. I have included in the result section as suggested.
Conclusions – author writes about purpose of the study not conclusions
I would like to thank the reviewer for this suggestion. The conclusion has been modified.
- Table 1
“Search results*”, what does symbol mean at the end of this phrase?
I have included a revised version of table 1 as requested.
“studies” is plural form, author should correct this everywhere where he writes “1 studies” to “1 study”
I would like to thank the reviewer for this suggestion. The manuscript has been checked for English errors.
- Table 3
Please unify the font
Done

Round 2
Reviewer 2 Report
I have no further remarks. Good luck with further processing of your manuscript.